# Genetic Differentiation among Livestock Breeds—Values for F_st_

**DOI:** 10.3390/ani12091115

**Published:** 2022-04-26

**Authors:** Stephen J. G. Hall

**Affiliations:** Department of Environmental Protection and Landscape, Estonian University of Life Sciences, Kreutzwaldi 5, 51014 Tartu, Estonia; s.hall973@btinternet.com

**Keywords:** livestock biodiversity, microsatellites, single-nucleotide polymorphisms, cattle, sheep, pigs, goats, horses, chickens

## Abstract

**Simple Summary:**

The degree of relationship among livestock breeds can be quantified by the F_st_ statistic, which measures the extent of genetic differentiation between them. An F_st_ value of 0.1 has often been taken as indicating that two breeds are indeed genetically distinct, but this concept has not been evaluated critically. Here, F_st_ values have been collated for the six major livestock species: cattle, sheep, goats, pigs, horses, and chickens. These values are remarkably variable both within and between species, demonstrating that F_st_ > 0.1 is not a reliable criterion for breed distinctiveness. However, the large body of F_st_ data accumulated in the last 20–30 years represents an untapped database that could contribute to the development of interdisciplinary research involving livestock breeds.

**Abstract:**

(1) Background: The F_st_ statistic is widely used to characterize between-breed relationships. F_st_ = 0.1 has frequently been taken as indicating genetic distinctiveness between breeds. This study investigates whether this is justified. (2) Methods: A database was created of 35,080 breed pairs and their corresponding F_st_ values, deduced from microsatellite and SNP studies covering cattle, sheep, goats, pigs, horses, and chickens. Overall, 6560 (19%) of breed pairs were between breeds located in the same country, 7395 (21%) between breeds of different countries within the same region, 20,563 (59%) between breeds located far apart, and 562 (1%) between a breed and the supposed wild ancestor of the species. (3) Results: General values for between-breed F_st_ were as follows, cattle: microsatellite 0.06–0.12, SNP 0.08–0.15; sheep: microsatellite 0.06–0.10, SNP 0.06–0.17; horses: microsatellite 0.04–0.11, SNP 0.08–0.12; goats: microsatellite 0.04–0.14, SNP 0.08–0.16; pigs: microsatellite 0.06–0.27, SNP 0.15–0.22; chickens: microsatellite 0.05–0.28, SNP 0.08–0.26. (4) Conclusions: (1) Large amounts of F_st_ data are available for a substantial proportion of the world’s livestock breeds, (2) the value for between-breed F_st_ of 0.1 is not appropriate owing to its considerable variability, and (3) accumulated F_st_ data may have value for interdisciplinary research.

## 1. Introduction

Much research effort over the last 30 years has been applied to the characterization of livestock breeds by molecular genetics, primarily by microsatellite (MS) and single-nucleotide polymorphism (SNP) technologies. This research has usually aimed to support the conservation and sustainable utilization of livestock biodiversity, and also to elucidate the processes of domestication and the evolution and differentiation of breeds.

One of the outputs has been the calculation of the extents to which breeds have diverged from each other, and a very widely used measure for this genetic differentiation has been the F_st_ statistic. As originally described [1], F_st_ values from 0.05 to 0.15 were taken to indicate moderate differentiation between populations, from 0.15 to 0.25 is high differentiation, and greater than 0.25 is very high differentiation. In principle, F_st_ could therefore be used to inform discussion relating to particular breeds, for example, that they are sufficiently different from each other to justify support for their conservation, or, conversely, that they are sufficiently similar for them to merge. In practice, F_st_ measurements are not often used as the main genetic justification for policy decisions regarding breed conservation, but the large number of F_st_ measurements available represents a data resource that could yield insights into overall patterns of breed differentiation. Indeed, genetic differentiation of breeds has often been placed in a spatial context by investigating how it is paralleled by geographic distance [2,3]. Further work has shown correlations with human [4,5] and ecological [6] diversity.

For many years, the literature has included such statements as “… the level often found between related breeds (e.g., F_st_ > 0.1) …” [7]; “a close relationship between [two breeds] (F_st_ = 0.019)…” [8]; “a threshold value …” [9]; “… strongly indicated that the two … are sufficiently different to be considered separate breeds” [10]; “… the overall differentiation assessed in the entire dataset was higher than most other studies carried out on European cattle …..” [11]; “… pairwise comparison … showed F_st_ < 0.1 and suggested clearly differentiated populations …” [12]. The present study aimed to provide an extensive review of the literature, and is therefore a test of the informal hypothesis embodied in the foregoing statements; namely, that differentiation of breeds can be signalled by F_st_ > 0.1.

## 2. Materials and Methods

Published data on F_st_ calculations were obtained from MS or SNP studies on cattle, sheep, goats, horses, pigs, sheep and chickens. A keyword search was not made because F_st_ is seldom used as a keyword or included in the title of a paper. The search proceeded initially by studying reference lists and citations of key papers such as [13,14,15,16,17] and, for cattle, an extensive bibliography assembled for a compilation on world cattle breeds [18]. Data presented solely as heatmaps or as Nei’s genetic distance were not used. Attempts were made to obtain unpublished information directly from authors. Only data that clearly used Wright’s F_st_ [19] were used, and Reynolds genetic distance measures D_R_ were transformed to F_st_ [20]. F_st_ calculations among herds or flocks were not used except when they related to differentiation between these entities and other, distinct breeds. Breed names and country affiliations were according to [18] when these were available, otherwise the usage considered most widespread and valid was adopted, or a breed name was assigned for the purposes of the study. Technical details such as sample sizes, numbers of alleles, and details of SNP technology were not considered. The references cited are listed in Table 1.

Preliminary analysis when 30,000 breed pairs had been obtained showed interpretable patterns of distribution of F_st_ for each of the twelve combinations of species and methodology (MS and SNP). Attention was then focused on recent publications, and a further 5080 breed pairs were added from a final total of 166 papers. No claim is made that this is a complete literature survey.

F_st_ calculations were classified according to whether the two breeds involved were affiliated to the same country, or to different countries. Those of different countries were coded according to the spatial relationships of the two countries, as defined by their borders. Pairs that included a wild ancestor (as defined in the respective studies) were also considered (Table 2).

For some analyses, to achieve an overview of coverage of different spatial relationships, F_st_ values relating to wild ancestor were excluded, and those for breed pairs classified as 2-Land-adj, 3-Marine-adj, 4-Nbut1, 5-Nbut1marine were merged into a combined geographical class designated Regional.

Some breeds, occurring internationally and often with national prefixes, were identified here as global breeds (Table 3) regardless of their country affiliation. No sheep breeds were thus designated. Although Merino sheep, for example, are very widely distributed, these populations represent well-established distinct breeds [178] and there is no equivalent to the global trades in germplasm seen, for example, in Holstein cattle, Large White (Yorkshire) pigs, and Angora goats. All breed pairs that included a global breed were classified as 6-Remote.

Owing to the large number of breeds considered, direct assessment of which breeds had been characterized by which methodology was not practicable, but preliminary analysis suggested that global breeds were more frequently included in studies using SNP approaches than in those using MS. This was investigated by comparing–for each methodology × species combination–the frequencies of occurrence of breed pairs, which included a global breed.

Statistical comparisons used non-parametric tests; specifically, the Kruskal-Wallis test and, for comparison of rank orders, Kendall’s coefficient of concordance [179,180,181].

## 3. Results

The literature search concluded in August 2021, with 35,080 F_st_ calculations having been assembled (Table 4). Numbers of breed pairs ranged very widely between studies, from 1 to 10,296.

The breakdown of the dataset according to spatial relationship is in Table 5, and according to specific breeds in Appendix A. The complete dataset is in Appendix A. In order to characterize the range of F_st_ values within each species × methodology group, the largest and smallest of the medians calculated for each spatial relationship were identified. The medians relating to wild ancestors were excluded for this purpose.

Species varied in the degree to which different spatial relationships were covered in the literature. Reflecting the relatively small numbers of breed pairs in the four regional classes (Table 2), in Table 6 these were condensed into geographical classes 1-Same, 6-Remote and Regional. Of the F_st_ calculations, 562 included a wild ancestor, and of the remaining 34,518, 59% (20,563) were of breed pairs classified as 6-Remote, 21% (7395) were Regional, and 19% (6560) were 1-Same. The proportion of breed pairs defined as 6-Remote was, for most species, higher in studies conducted with SNP methodologies than in MS studies.

F_st_ values for breed pairs also varied according to the spatial relationships of breeds. In all twelve (species × methodology) cases from cattle MS through to chicken SNP, differences in F_st_ between spatial relationships were highly significant (*p* < 0.001; Kruskal-Wallis statistic, d.f. in brackets, respectively, 201.59 (5), 1165.5 (5), 186.71 (6), 1117.9 (6), 914.75 (6), 19.39 (5), 249.67 (5), 1605.03 (6), 207.23 (6), 120.45 (6), 196.11 (6), 107.11 (6)).

The rank orders of the median F_st_ values for each of the spatial relationship categories (excluding 7-Wild_ancestor) were significantly correlated (Kendall concordance test; for MS, W = 0.52, for SNP, W = 0.66, both *p* < 0.01).

These differences are illustrated in Figure 1 and Figure 2, for MS and SNP data, respectively.

## 4. Discussion

It is reported [182] that there are 5517 livestock breeds in the world (1047 cattle, 1164 sheep, 580 goat, 720 horse, 569 pig, and 1437 chicken). It is evident that about one-fifth of the world’s breeds are represented in the dataset assembled in this report; at least 1040 different breeds have been studied by MS, and 797 by SNP (both methodologies have been applied to some breeds, almost always in separate studies), respectively. Principal reasons for this work have included characterization and conservation of this livestock biodiversity. Much of it has been on establishing the extent of differentiation of national breeds from those of remote countries (many of which are global breeds), with an emphasis on breed pairs of which one member was a national breed and the other was from a remote country (59% of breed pairs). Only 19% of breed pairs comprised breeds that were both of the same country. Thus, an unexpected result of this study has been to suggest that so far, as conservation is concerned, genetic studies have been more interested in introgression of breeds from abroad, than in maintaining the genetic distinctiveness of the diverse breeds of a country. This tendency is evident in both MS and SNP studies, particularly the latter. However, as 21% of breed pairs related to breeds of different–but nearby–countries, there has been a degree of interest in regional patterns of breed differentiation.

The original aim of this study was, however, to test the general prediction that a realistic threshold value for between-breed F_st_ is 0.1. The ranges of F_st_ values between pairs of breeds are shown to be so wide that this prediction appears obsolete for practical purposes. It is now very evident that the F_st_ approach is only one method of visualizing the findings of genomic studies of breeds [183], and reports are now typically accompanied by genetic distance calculations, STRUCTURE plots, plots generated by multivariate statistics, and heatmaps, often within a framework of landscape genomics [12]. Nevertheless, there may still be a requirement for benchmark values of F_st_ as indicating breed differentiation, for example for interdisciplinary studies or to provide a context for conservation genetics of wild populations. For these purposes, the following benchmarks could be adopted, based on the median values obtained in the present study, cattle: MS 0.06–0.12, SNP 0.08–0.15; sheep: MS 0.06–0.10, SNP 0.06–0.17; horse: MS 0.04–0.11, SNP 0.08–0.12; goat: MS 0.04–0.14, SNP 0.08–0.16; pig: MS 0.06–0.27, SNP 0.15–0.22; chicken: MS 0.05–0.28, SNP 0.08–0.26. However, use of these values as benchmarks must be conditional on acknowledging their stochastic nature and probable dependence on geographical factors.

The finding that different spatial relationships of breed pairs may influence F_st_ values is novel but not surprising. F_st_ statistics are well known [10,17,184] to lead to insights into patterns of migration and gene flow when placed in a geographical framework. In principle, a formal statistical analysis of the dataset assembled for this paper might enable quantification of the relative contributions of the different variates (to include species, methodology, and spatial relationship) but with the public availability of genotype data, the extensive meta-analysis of published F_st_ values from earlier studies may itself be an obsolete approach, as raw genotypes from multiple sources could be combined and F_st_ values reliably calculated from the pooled data.

The considerable amount of F_st_ data accumulated over the last few decades is still likely to represent a valuable resource. It could be used to audit breed conservation activities, although it will not be a definitive determinant of whether a breed is truly distinctive [185,186]. At the level of original research, these data may help in the formation of hypotheses for future work on breed differentiation and, as awareness increases of their existence and accessibility, they could provide stimulus for new interdisciplinary research.

## 5. Conclusions

The use of specific values of F_st_ as indicating breed differentiation is not justified, but benchmark values can be proposed for use in specified contexts. F_st_ data, as obtained from published studies, represent a resource for interdisciplinary research.

## Figures and Tables

**Figure 1 animals-12-01115-f001:**
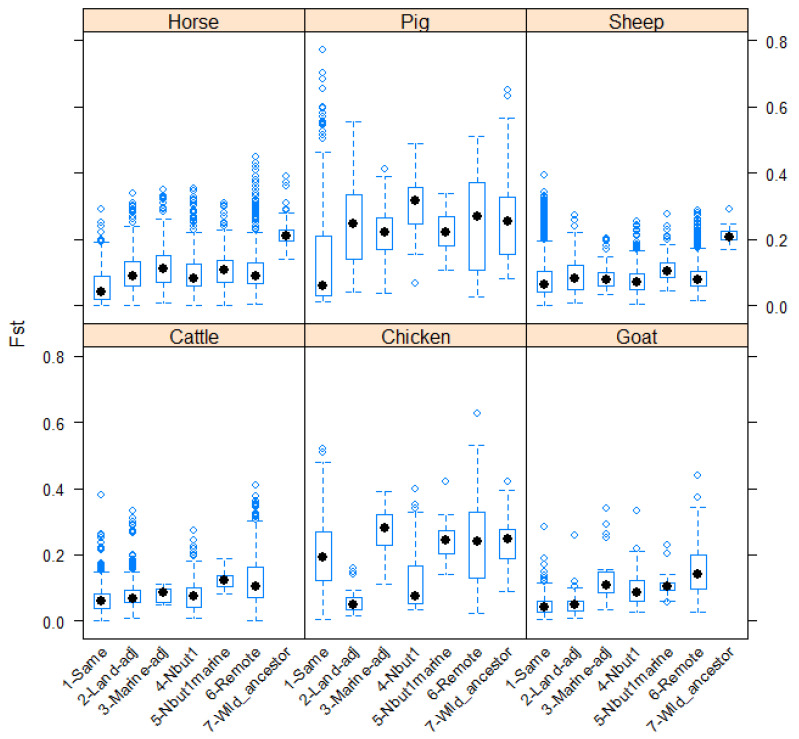
Boxplots summarizing, for each species, values of the F_st_ statistic for breed pairs according to the spatial relationship of the members of the pair. Data from studies using MS methodologies.

**Figure 2 animals-12-01115-f002:**
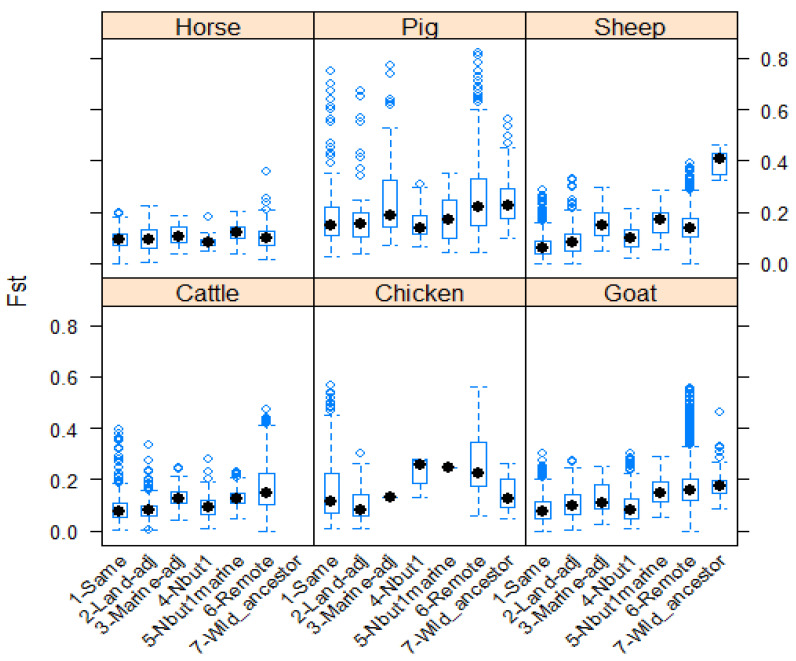
Boxplots summarizing, for each species, values of the F_st_ statistic for breed pairs according to the spatial relationship of the members of the pair. Data from studies using SNP methodologies.

**Table 1 animals-12-01115-t001:** Data sources. MS: microsatellites; SNP: single-nucleotide polymorphisms. The reference codes are internal to the project and enable cross-reference to Appendix A.

Species	Methodology	Reference Code and Citation	Number of Breed Pair F_st_ Calculations
Cattle	MS	11974 [2]	2
Cattle	MS	12398 [21]	21
Cattle	MS	12500 [22]	2
Cattle	MS	12919 [23]	21
Cattle	MS	13438 [20]	153
Cattle	MS	13930 [24]	3
Cattle	MS	14407 [25]	6
Cattle	MS	18988 [26]	10
Cattle	MS	18989 [27]	153
Cattle	MS	19230 [28]	6
Cattle	MS	19287 [29]	21
Cattle	MS	1929 [30]	6
Cattle	MS	19335 [31]	595
Cattle	MS	19649 [32]	10
Cattle	MS	19651 [33]	6
Cattle	MS	19652 [34]	36
Cattle	MS	19653 [35]	21
Cattle	MS	7445 [36]	209
Cattle	MS	8012 [37]	10
Cattle	SNP	11390 [38]	1081
Cattle	SNP	11637 [39]	28
Cattle	SNP	16251 [40]	21
Cattle	SNP	13194 [41]	15
Cattle	SNP	16089 [42]	15
Cattle	SNP	18750 [4]	105
Cattle	SNP	18783 [43]	496
Cattle	SNP	18993 [44]	741
Cattle	SNP	19039 [45]	378
Cattle	SNP	19104 [46]	36
Cattle	SNP	19158 [47]	135
Cattle	SNP	19261 [48]	435
Cattle	SNP	19277 [49]	55
Cattle	SNP	19287 [29]	21
Cattle	SNP	19409 [50]	210
Chicken	MS	17999 [51]	15
Chicken	MS	19057 [16]	276
Chicken	MS	19163 [52]	21
Chicken	MS	19166 [53]	59
Chicken	MS	19248 [54]	120
Chicken	MS	19280 [55]	10
Chicken	MS	19281 [56]	78
Chicken	MS	19310 [57]	10
Chicken	MS	19330 [58]	36
Chicken	MS	19331 [59]	28
Chicken	MS	19414 [60]	3
Chicken	MS	19415 [61]	105
Chicken	MS	19416 [62]	15
Chicken	MS	19434 [63]	28
Chicken	MS	19637 [64]	288
Chicken	MS	19642 [65]	15
Chicken	MS	19654 [66]	28
Chicken	MS	19655 [67]	1
Chicken	MS	19656 [68]	45
Chicken	SNP	19192 [69]	21
Chicken	SNP	19283 [70]	28
Chicken	SNP	19302 [71]	45
Chicken	SNP	19316 [72]	105
Chicken	SNP	19331 [59]	28
Chicken	SNP	19431 [73]	231
Chicken	SNP	19432 [74]	28
Chicken	SNP	19433 [75]	21
Chicken	SNP	19533 [76]	28
Goat	MS	16043 [77]	15
Goat	MS	18124 [78]	36
Goat	MS	19107 [79]	21
Goat	MS	19108 [80]	3
Goat	MS	19109 [81]	10
Goat	MS	19110 [82]	10
Goat	MS	19122 [83]	155
Goat	MS	19125 [84]	66
Goat	MS	19215 [85]	4
Goat	MS	19231 [86]	6
Goat	MS	19393 [87]	36
Goat	MS	19601 [88]	10
Goat	MS	19603 [89]	36
Goat	MS	19605 [90]	10
Goat	MS	19606 [91]	21
Goat	MS	19609 [92]	3
Goat	MS	19611 [93]	21
Goat	MS	19646 [94]	45
Goat	SNP	15770 [95]	105
Goat	SNP	18126 [17]	10,296
Goat	SNP	19205 [96]	136
Goat	SNP	19256 [97]	28
Goat	SNP	19305 [98]	3
Goat	SNP	19643 [99]	21
Horse	MS	13437 [100]	28
Horse	MS	15788 [101]	55
Horse	MS	19025 [102]	7
Horse	MS	19162 [13]	701
Horse	MS	19169 [103]	136
Horse	MS	19199 [104]	325
Horse	MS	19211 [105]	21
Horse	MS	19264 [106]	4371
Horse	MS	19300 [107]	1
Horse	MS	19303 [108]	902
Horse	MS	19668 [109]	3
Horse	SNP	16649 [110]	665
Horse	SNP	19260 [111]	36
Horse	SNP	19265 [112]	3
Horse	SNP	19299 [113]	3
Horse	SNP	19300 [107]	1
Horse	SNP	19301 [114]	15
Horse	SNP	19400 [115]	20
Pig	MS	1144 [116]	21
Pig	MS	12225 [117]	15
Pig	MS	19293 [118]	28
Pig	MS	19295 [119]	136
Pig	MS	19327 [120]	351
Pig	MS	19426 [121]	21
Pig	MS	19636 [122]	3
Pig	MS	19639 [123]	28
Pig	MS	554 [124]	231
Pig	MS	6025 [125]	45
Pig	MS	8242 [126]	36
Pig	SNP	19058 [127]	91
Pig	SNP	19234 [128]	6
Pig	SNP	19262 [129]	3
Pig	SNP	19270 [130]	136
Pig	SNP	19272 [131]	105
Pig	SNP	19276 [132]	21
Pig	SNP	19292 [133]	21
Pig	SNP	19328 [134]	28
Pig	SNP	19410 [135]	171
Pig	SNP	19412 [136]	253
Pig	SNP	19420 [137]	55
Pig	SNP	19422 [138]	45
Pig	SNP	19425 [139]	3
Pig	SNP	19661 [140]	66
Sheep	MS	19244 [141]	45
Sheep	MS	14033 [142]	3
Sheep	MS	15852 [143]	153
Sheep	MS	17623 [144]	21
Sheep	MS	19002 [145]	117
Sheep	MS	19032 [146]	12
Sheep	MS	19221 [147]	55
Sheep	MS	19222 [148]	21
Sheep	MS	19223 [149]	36
Sheep	MS	19239 [8]	91
Sheep	MS	19240 [150]	55
Sheep	MS	19243 [151]	66
Sheep	MS	19246 [152]	10
Sheep	MS	19247 [153]	36
Sheep	MS	19287 [29]	146
Sheep	MS	19381 [154]	15
Sheep	MS	19385 [155]	55
Sheep	MS	19597 [156]	66
Sheep	MS	19598 [157]	325
Sheep	MS	19599 [158]	15
Sheep	MS	19604 [159]	10
Sheep	MS	19625 [160]	276
Sheep	MS	19630 [161]	15
Sheep	MS	2473 [162]	190
Sheep	MS	4401 [9]	249
Sheep	MS	4722 [14]	1596
Sheep	SNP	18433 [163]	55
Sheep	SNP	18991 [164]	3403
Sheep	SNP	19094 [165]	300
Sheep	SNP	19132 [166]	89
Sheep	SNP	19136 [167]	105
Sheep	SNP	19154 [168]	15
Sheep	SNP	19214 [169]	78
Sheep	SNP	19226 [170]	3
Sheep	SNP	19296 [171]	6
Sheep	SNP	19332 [172]	3
Sheep	SNP	19436 [173]	153
Sheep	SNP	19447 [174]	27
Sheep	SNP	19499 [175]	45
Sheep	SNP	19561 [176]	21
Sheep	SNP	19648 [177]	10

**Table 2 animals-12-01115-t002:** Classification of geographical relationships of breed pairs.

Spatial Relationship of Breeds	Code	Geographical Class
In the same country	1-Same	Same country
In countries sharing a land border	2-Land-adj	
In countries sharing a water border (sea or lake)	3-Marine-adj	
In countries separated by a third country with land borders	4-Nbut1	Regional
In countries separated by a third country with a water border	5-Nbut1marine	
In more widely separated countries	6-Remote	Remote
One member of breed pair a wild ancestor ^(1)^	7-Wild_ancestor	Wild ancestor

^(1)^ Mouflon, bezoar, wild boar, red jungle fowl, Przewalski horse.

**Table 3 animals-12-01115-t003:** Global breeds.

Cattle	Chicken	Goat	Horse	Pig
Aberdeen Angus	AVIANDIV ^(1)^	Alpine	Thoroughbred/Pur Sang	Large White
Brown Swiss	Commercial ^(2)^	Angora		Piétrain
Charolais	Rhode Island Red	Nubian/Anglo Nubian		
Holstein		Saanen		
Guernsey		Toggenburg		
Hereford				
Jersey				
Limousin				
Simmental				

^(1)^ F_st_ calculations from a global panel of commercial chicken breeds [56]. ^(2)^ All commercial breeds, varieties, and strains.

**Table 4 animals-12-01115-t004:** Summary of studies from which F_st_ data were extracted, according to species and methodology.

	Cattle	Sheep	Goat	Horse	Pig	Chicken	Totals
MS							
Number of studies	19	26	18	11	11	19	104
Total number of breeds	143	302	124	190	98	183	1040
Median (range) breed pairs per study	10 (2–595)	55 (3–1596)	18 (3–155)	55 (1–4371)	28 (3–351)	28 (1–288)	
Total breed pairs	1291	3679	508	6550	915	1181	14,124
SNP							
Number of studies	15	15	6	7	14	9	66
Total number of breeds	184	192	176	61	106	78	797
Median (range) breed pairs per study	105 (15–1081)	45 (3–3403)	67 (3–10,296)	15 (1–665)	50 (3–253)	28 (21–231)	
Total breed pairs	3772	4313	10,589	743	1004	535	20,906

**Table 5 animals-12-01115-t005:** Summary statistics for F_st_ between breed pairs, classified by method (MS and SNP) and by spatial relationships between breeds. For definitions as abbreviated here, see Table 2.

Species and Methodology	Median	Max	Min	Mean	Breed Pairs
Cattle MS					
1-Same	0.06 ^(1)^	0.381	0.001	0.069	320
2-Land-Adj	0.069	0.331	0.007	0.083	260
3-Marine-adj	0.085	0.113	0.048	0.078	11
4-Nbut1	0.074	0.273	0.009	0.082	144
5-Nbut1marine	0.124	0.19	0.081	0.128	9
6-Remote	0.105	0.408	0.001	0.127	547
Cattle SNP	Median	Max	Min	Mean	breed pairs
1-Same	0.077	0.394	0.002	0.087	976
2-Land-Adj	0.08	0.335	0.002	0.085	604
3-Marine-adj	0.127	0.246	0.045	0.13	201
4-Nbut1	0.092	0.279	0.01	0.095	165
5-Nbut1marine	0.126	0.231	0.047	0.127	271
6-Remote	0.148	0.474	0.001	0.172	1555
Sheep MS	Median	Max	Min	Mean	breed pairs
1-Same	0.063	0.395	0	0.084	1258
2-Land-Adj	0.081	0.271	0.008	0.089	403
3-Marine-adj	0.079	0.204	0.034	0.084	181
4-Nbut1	0.071	0.253	0.006	0.079	416
5-Nbut1marine	0.102	0.275	0.046	0.112	120
6-Remote	0.078	0.288	0.014	0.089	1290
7-Wild_ancestor	0.206	0.291	0.171	0.213	11
Sheep SNP	Median	Max	Min	Mean	breed pairs
1-Same	0.061	0.286	0	0.072	643
2-Land-Adj	0.083	0.328	0	0.09	343
3-Marine-adj	0.146	0.3	0.049	0.155	188
4-Nbut1	0.096	0.217	0.024	0.102	160
5-Nbut1marine	0.171	0.288	0.054	0.166	132
6-Remote	0.136	0.391	0	0.145	2833
7-Wild_ancestor	0.411	0.462	0.327	0.397	14
Horse MS	Median	Max	Min	Mean	breed pairs
1-Same	0.041	0.291	0	0.058	763
2-Land-Adj	0.09	0.339	0	0.103	515
3-Marine-adj	0.11	0.349	0.007	0.119	244
4-Nbut1	0.08	0.354	0	0.101	372
5-Nbut1marine	0.106	0.31	0	0.109	546
6-Remote	0.09	0.45	0.003	0.105	3938
7-Wild_ancestor	0.21	0.389	0.14	0.217	172
Horse SNP	Median	Max	Min	Mean	breed pairs
1-Same	0.091	0.201	0.002	0.095	69
2-Land-Adj	0.094	0.227	0.006	0.099	50
3-Marine-adj	0.107	0.185	0.038	0.111	31
4-Nbut1	0.084	0.183	0.051	0.09	19
5-Nbut1marine	0.121	0.202	0.04	0.119	59
6-Remote	0.098	0.36	0.015	0.107	515
Goat MS	Median	Max	Min	Mean	breed pairs
1-Same	0.042	0.283	0.005	0.05	208
2-Land-Adj	0.048	0.259	0.01	0.055	40
3-Marine-adj	0.107	0.339	0.035	0.136	19
4-Nbut1	0.085	0.331	0.027	0.105	40
5-Nbut1marine	0.104	0.229	0.058	0.111	24
6-Remote	0.141	0.439	0.026	0.153	177
Goat SNP	Median	Max	Min	Mean	breed pairs
1-Same	0.076	0.303	0	0.084	670
2-Land-Adj	0.096	0.274	0.005	0.103	550
3-Marine-adj	0.108	0.252	0.028	0.13	85
4-Nbut1	0.084	0.303	0.094	0.096	440
5-Nbut1marine	0.15	0.293	0.054	0.16	138
6-Remote	0.161	0.556	0.166	0.176	8547
7-Wild_ancestor	0.172	0.464	0.179	0.179	159
Pig MS	Median	Max	Min	Mean	breed pairs
1-Same	0.06	0.774	0.01	0.134	464
2-Land-Adj	0.247	0.556	0.039	0.244	74
3-Marine-adj	0.221	0.413	0.036	0.214	28
4-Nbut1	0.317	0.49	0.067	0.303	12
5-Nbut1marine	0.22	0.34	0.108	0.224	43
6-Remote	0.269	0.512	0.025	0.249	240
7-Wild_ancestor	0.254	0.65	0.081	0.269	54
Pig SNP	Median	Max	Min	Mean	breed pairs
1-Same	0.15	0.75	0.03	0.184	299
2-Land-Adj	0.156	0.67	0.04	0.185	72
3-Marine-adj	0.19	0.77	0.071	0.268	52
4-Nbut1	0.139	0.31	0.067	0.152	70
5-Nbut1marine	0.17	0.35	0.043	0.179	21
6-Remote	0.22	0.82	0.046	0.263	417
7-Wild_ancestor	0.23	0.56	0.1	0.243	73
Chicken MS	Median	Max	Min	Mean	breed pairs
1-Same	0.194	0.52	0.006	0.198	629
2-Land-Adj	0.048	0.16	0.017	0.061	35
3-Marine-adj	0.28	0.39	0.11	0.269	25
4-Nbut1	0.075	0.4	0.036	0.115	139
5-Nbut1marine	0.245	0.42	0.14	0.25	16
6-Remote	0.24	0.63	0.022	0.242	300
7-Wild_ancestor	0.249	0.12	0.088	0.251	37
Chicken SNP	Median	Max	Min	Mean	breed pairs
1-Same	0.113	0.565	0.011	0.166	261
2-Land-Adj	0.08	0.304	0.007	0.116	22
3-Marine-adj	0.132	0.132	0.132	0.132	1
4-Nbut1	0.255	0.28	0.13	0.23	4
5-Nbut1marine	0.245	0.245	0.245	0.245	1
6-Remote	0.226	0.56	0.06	0.259	204
7-Wild_ancestor	0.125	0.261	0.049	0.141	42

^(1)^ For each species × methodology combination the greatest and lowest median values are underlined (excluding those relating to wild ancestors).

**Table 6 animals-12-01115-t006:** Proportions of breed pairs in each geographical class, compared between methodologies.

	MS	SNP	MS	SNP	MS	SNP	MS	SNP
	Total	Total	% 1-Same	% 1-Same	% Regional	% Regional	% 6-Remote	% 6-Remote
Cattle	1291	3772	24.8	25.9	32.8	32.9	42.4	41.2
Chicken	1144	493	55.0	52.9	18.8	5.7	26.2	41.4
Goat	508	10,430	40.9	6.4	24.2	11.6	34.8	81.9
Horse	6378	743	12.0	9.3	26.3	21.4	61.7	69.3
Pig	861	931	53.9	32.1	18.2	23.1	27.9	44.8
Sheep	3668	4299	34.3	15.0	30.5	19.1	35.2	65.9

## Data Availability

All data are available in Appendix A.

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
