# Peer review of "Genetic Differentiation among Livestock Breeds—Values for Fst"

_animals, 2022, doi:10.3390/ani12091115_

Round 1

Reviewer 1 Report

This was a very interesting paper and the data collection, collation and analysis impressive. A very large undertaking. The data in the supplementary files provides a valuable resource (and infrastructure for future data collation) and adds to the transparency of the results.

I have attached a pdf with a few minor suggestions and edits. 

Reviewer 2 Report

The Author provided a review of literature on Fst statistic widely used to characterize between-breed relationships. The object of the article is certainly of interest.

The manuscript analyzed a large database of Fst values deduced from microsatellite and SNP studies covering cattle, sheep, goat, pig, horse and chicken. The Author tested also, the hypothesis that some values of Fst in relation to differentiation of breeds can be considered typical.

Here are some suggestions that the Author can consider improving the manuscript.

Materials and Methods

The statistical methods applied to the dataset are missing.

Results

Table 4: please explain how the Author obtain the “Median (range) breed pairs per study”. What is the meaning of this data?

Lines 107-109: please explain the significance of this analysis.

Table 5: please change “…..abbreviated here, see Table 1.” with “…..abbreviated here, see Table 2.”

Table 6: please enlarge and center the columns.

Lines 124-130: The statistical methods applied have to be reported in MM.

Discussion

Lines 145-149: I disagree with the Author. The conservation of genetic identity of local populations and the introgression of other breeds, often global breeds used to improve performance of local breeds, are two complementary and closely related aspects. This aspect should be further discussed. Furthermore, the difference between 19% and 21% is very small.

Lines 155-158: The benchmark values of Fst proposed are based on median values obtained by spatial relationships between breeds. Please specify.

Reviewer 3 Report

Comments are attached for authors
